# Sarcopenia among the Elderly Population: A Systematic Review and Meta-Analysis of Randomized Controlled Trials

**DOI:** 10.3390/healthcare9060650

**Published:** 2021-05-31

**Authors:** Di-Ya Tu, Fa-Min Kao, Shih-Tzer Tsai, Tao-Hsin Tung

**Affiliations:** 1Department of Nutrition Therapy, Cheng-Hsin General Hospital, Taipei 112, Taiwan; snoopy5386@hotmail.com (D.-Y.T.); felix19940728@gmail.com (F.-M.K.); tsaist60@gmail.com (S.-T.T.); 2Department of Internal Medicine, Cheng-Hsin General Hospital, Taipei 112, Taiwan; 3Evidence-Based Medicine Center, Taizhou Hospital of Zhejiang Province Affiliated to Wenzhou Medical University, Linhai 317000, China

**Keywords:** protein supplement, sarcopenia, elderly population, meta-analysis

## Abstract

*Purpose.* This systematic review and meta-analysis was conducted to explore the effect of protein intake on the prevention and improvement of sarcopenia. *Methods.* We searched the Cochrane Library, PubMed, and EMBASE from inception to 20 May 2021. Two authors independently selected studies, assessed the quality of included studies, and extracted data. Any disagreement was resolved by discussion with a third author. *Results.* There were 12 studies that met the selection criteria among 53 eligible publications. The results of the study show that the protein intake has no significant effect on the physical performance—4 m gait speed, chair rise test, short physical performance battery, muscle mass—skeletal muscle mass index, and muscle strength—hand grip strength. *Conclusion.* Protein supplementation had no significant effect on 4 m gait speed and on improving skeletal muscle mass index, hand grip strength, chair rise test, and short physical performance battery. Additional randomized controlled trials are warranted to adequately assess the effect of protein supplementation on elderly sarcopenia.

## 1. Introduction

Sarcopenia is a major health issue among the older population due to the increased risk of adverse outcomes, including falls, frailty, disability, morbidity, and mortality [1]. Sarcopenia is a symptom of skeletal muscle mass and strength loss, and physical performance decline [2]. There is a significant change in body composition with age, such as a decline in muscle mass and gain in body fat. The muscle mass reduces with age, by about 1/3 in those over 50 years old and by a further 15% in those between 70 and 80 years old [3]. On average, the estimated prevalence of sarcopenia in the elderly—those aged 60–70 years—is 5–13%, and increases to 11–50% in those aged over 80 years [4].

From the nutrition viewpoint, protein intake in the diet is a key factor to know whether sarcopenia has happened or not. Protein is an important component of cells in the body. The human body uses amino acids to perform a number of metabolic functions, such as acting as enzymes, membrane carriers, and hormones [5]. Lack of protein in the body easily leads to frailty, as well as impaired immune function and wound healing [6]. The recommended dietary allowance of Taiwan for men and women aged 50–70 years is set at 55 g/day and 50 g/day, respectively, and for those aged above 71 years old, it is set at 60 g/day for men and 50 g/day for women [7]. More protein intake is recommended for the elderly due to age-related loss in total body protein, resulting in an increased risk of sarcopenia.

In recent years, many studies have indicated the effect of protein intake on sarcopenia through different types of protein such as soy protein, milk protein, whey protein, leucine, branched-chain amino acid, and β-hydroxy-β-methylbutyric acid. However, the effect of protein intake on sarcopenia among the elderly population is still not clear. This systematic review and meta-analysis was conducted to explore the effect of protein intake on the prevention and improvement of sarcopenia.

## 2. Materials and Methods

### 2.1. Literature Search

In this study, we conducted a systematic review and meta-analysis to assess the effect of protein intake on sarcopenia among the elderly population. We used the PICO framework to search the terms (P: Elderly OR Elder OR Older OR Aged) AND (I: Protein intake OR Leucine OR HMB OR β-hydroxy-β-methylbutyric acid OR β-hydroxy-β-methylbutyrate) AND (O: Muscle mass OR Muscle strength OR Hand grip strength OR Gait speed OR Chair stand test OR Short physical performance battery) in the following databases: PubMed, Embase, and the Cochrane Central Register of Controlled Trials from inception through 20 May 2021, for relevant publications without limitations in the English language. 

This systematic review was carried out in accordance with the Preferred Reporting Items for Systematic Reviews and Meta-Analysis (PRISMA^®^) statement guidelines. Figure 1 shows the process of the study selection. The search strategy is detailed in Table 1. The study protocol was registered on PROSPERO (No. 198995).

### 2.2. Study Selection

The included studies met the following inclusion criteria: the study design was a randomized controlled trial (RCT), participants were human, the experimental group received protein, and the control group received a placebo. Firstly, we discarded duplicate studies by screening titles and abstracts. Secondly, in accordance with the previously designed study inclusion criteria, eligible studies were extracted by reviewing full texts. The titles and abstracts of all studies identified by our search were independently assessed by two of the authors (Tu and Kao) for eligibility. These authors checked the full text of potentially eligible trials to determine whether they met the inclusion criteria. The third author (Tung) arbitrated when the two authors disagreed on the inclusion of a study.

### 2.3. Data Extraction and Assessment of Potential Bias

We performed the data extraction and risk-of-bias assessment process. For all articles included, the following characteristics were obtained: first author, publication year, study country, participants in the RCTs, characteristics of intervention and comparison groups, and main outcome measurements. We reviewed the titles and abstracts when searching the relevant studies after all references had been imported to EndNote^®^ software. After a thorough appraisal of these selected publications, we indexed the full texts and subsequently assessed the risk of bias using the *Cochrane Handbook for Systematic Reviews of Interventions* [8]. The handbook includes seven domains of bias risk: (1) random sequence generation, (2) allocation concealment, (3) blinding of participants and personnel, (4) blinding of outcome assessment, (5) incomplete outcome data, (6) selective reporting, and (7) other sources of bias. We (Tu and Kao) used the Cochrane Collaboration tool to assess the risk of bias in the included trials [9]. Any disagreement was resolved through discussion with the third author (Tung). 

### 2.4. Statistical Analysis 

A conservative random-effects meta-analysis model was used throughout because of heterogeneity in study populations, interventions, and study designs. We used Review Manager version 5.4.1 to calculate the overall effect of protein intake on sarcopenia [10] for the elderly population. Heterogeneity in meta-analysis refers to variation in study outcomes between studies. In this study, we used the *I*^2^ and inconsistency statistics. The statistics describe the percentage of variation across studies that is caused by heterogeneity rather than chance [11]; values of <25%, between 25% and 75%, and >75% were considered low, moderate, and high heterogeneity, respectively. A 95% confidence interval (CI) was constructed through the iterative noncentral chi-square distribution method [12]. In addition, we applied a fixed-effect model when it was less than 50% and would have applied the random-effects model if it had been 50% or more. *p* < 0.05 indicated a significant difference between the protein intake and control groups. The results are presented as mean ± SD and 95% CIs. Forest plots were used to summarize results, and funnel plots were used to investigate publication bias.

## 3. Results

### 3.1. Baseline Characteristics of the Selected Studies

We searched 739 articles in total and sorted them with EndNote X7^®^. A total of 225 duplicates were excluded. We removed 461 studies for different reasons (reviews, meta-analyses, animal or cell experiments, and no relationship). Fifty-three papers were left for further assessment after reading the titles and abstracts. After reading the full articles, it was observed that 41 studies did not satisfy the selection criteria. Finally, 12 articles were included in the systematic review and meta-analysis [13,14,15,16,17,18,19,20,21,22,23,24]. A of 872 participants (440 and 432 participants in the intervention group and the controls) that met the eligibility criteria were included in this review (Table 2). Sample sizes ranged from 32 to 146.

### 3.2. Risk-of-Bias Assessment

We assessed the risk of bias by the *Cochrane Handbook for Systematic Reviews of Interventions* in the selected studies [8]. The results are shown in Figure 2. One study [21] was classed “high risk” and six studies [13,14,15,17,19,24] classed “unclear” with selection bias for allocation concealment. For the performance bias, Berton (2015) [14] classed “high risk” and Yamamoto (2021) [24] classed “unclear.” Three studies (Amasene, 2019; Berton, 2015; Mori, 2018) [13,14,18] were assessed as “high risk” and two studies [23,24] assessed as “unclear” with detection bias. For attrition bias, five studies (Amasene, 2019; Berton, 2015; Nilsson, 2020; Stout, 2013; Tieland, 2012) were judged “high risk” [13,14,19,21,23].

### 3.3. Outcome Measures

#### 3.3.1. The Effects of Protein Intake on Muscle Mass and Strength

Six trials with 235 and 235 participants in the intervention and control groups, respectively, assessed the effect of protein intake on muscle mass with a focus on the skeletal muscle mass index [14,15,17,18,19,20]. With random-effects models, there were no significant changes in skeletal muscle mass by protein intake (mean difference = 0.01, 95% CI: −0.07, 0.10). The *I*^2^ value was 54%, and the related *p* value was 0.74. The corresponding results are presented in Figure 3a. These data show that protein intake did not significantly affect skeletal muscle mass.

Ten RCTs with 383 and 381 participants in the intervention and control groups, respectively, assessed the effect of protein intake on muscle strength with a focus on hand grip strength [14,15,17,18,19,20,21,22,23,24]. With random-effects models, no difference in hand grip strength was observed in protein intake compared with control conditions (mean difference = 0.62, 95% CI: −0.37, 1.60). The *I*^2^ value was 93%, and the related *p* value was 0.22. The corresponding results are presented in Figure 3b. These data show that protein intake did not significantly affect hand grip strength.

#### 3.3.2. The Effect of Protein Intake on Physical Performance

##### 4 m Gait Speed

There were four trials with 140 and 136 participants in the intervention and control groups, respectively [14,19,22,23]. With fixed-effects models, protein intake did not have significant changes in 4 m gait speed (mean difference = −0.03, 95% CI: −0.06, 0.00). The *I*^2^ value was 0%, and the related *p* value was 0.05. The corresponding results are presented in Figure 3c. These data show that protein intake did not significantly affect 4 m gait speed.

##### Chair Rise Test

There were five trials with 204 and 199 participants in the intervention and control groups, respectively [14,19,20,22,23]. With random-effects models, there were no changes in the chair rise test (mean difference = 1.70, 95% CI: −1.54, 4.93). The *I*^2^ value was 99%, and the related *p* value was 0.30. The corresponding results are presented in Figure 3d. These data show that protein intake did not significantly affect the chair rise test.

##### Short Physical Performance Battery (SPPB)

There were seven trials with 276 and 267 participants in the intervention and control groups, respectively [13,14,15,16,19,20,23]. With random-effects models, there were no changes in SPPB (mean difference = 0.50, 95% CI: −0.25, 1.25). The *I*^2^ value was 95%, and the related *p* value was 0.19. The corresponding results are presented in Figure 3e. These data show that protein intake did not significantly affect SPPB.

### 3.4. Publication Bias

Publication bias is defined as the failure to publish the results of a study depending on the direction and statistical significance of the study findings [8]. As Figure 4 shows, the funnel plot of 4 m gait speed (c) is symmetric, but the funnel plots of skeletal muscle mass index (a), hand grip strength (b), chair rise test (d), and SPPB (e) are asymmetric and present some publication bias.

## 4. Discussion

### 4.1. Clinical Implication

The purpose of this meta-analysis was to know the effect of protein intake on the prevention and improvement of sarcopenia. The results of the study show that the protein intake had no significant effect on the physical performance—4 m gait speed, chair rise test, SPPB, muscle mass—skeletal muscle mass index, and muscle strength—hand grip strength.

Of the reviewed trials included, only two had all the parameters of sarcopenia diagnosis, such as muscle mass, muscle strength, and physical function, which were updated by the Asian Working Group in 2019 [25]; we were unable to explore the effect of protein intake in preventing and treating the outcome of sarcopenia completely. Furthermore, we need more studies that include all the outcomes in connection with the Asian Working Group, which recommended the outcome for sarcopenia diagnosis.

Protein intake enhances muscle protein synthesis; the elderly need sufficient protein intake to prevent the risk of sarcopenia. Many people consume protein supplements habitually, without considering body weight and the total protein intake from meals. According to the Sarcopenia and Physical Frailty in Older People: Multi-component Treatment Strategies (SPRINTT) project, the average daily protein intake of at least 1.0–1.2 g/kg body mass could prevent sarcopenia [26]. The safety and effectiveness concerning the consumption of 1.4 g of protein/kg of body weight (or more) in the older population has been studied [27]. Dietary protein provides amino acids, which are necessary for the synthesis of muscle protein. However, it is not clear whether protein requirements need to be higher in the older population to keep nitrogen balance and to prevent loss of muscle mass and strength [28].

In addition, the components of protein used in the intervention groups of the included trials varied. Although well-planned diet regimens may have clinical effectiveness, or are possibly better than individual nutrient supplements in maintaining muscle mass and physical function in elderly adults, we need more RCTs for further subgroup analysis to assess whether various protein sources are a confounding factor and then to confirm that improvement is caused by specific proteins and effective dosage in the elderly population. 

### 4.2. Practical Applications

It may be appropriate to assess the nutritional condition in elderly adults who are at risk of sarcopenia and frailty through investigations, body weight assessment, and selected blood compositions due to a sufficient nutritional profile that appears to help sustain muscle mass and preserve levels of physical function. From the clinical viewpoint, to prevent and improve sarcopenia, the strategy of protein supplements needs to consider body weight and protein intake from meals. The registered dietician will take a complete nutrition assessment, including anthropometry, biochemical data, clinical findings, and dietary recall to calculate the dosage of protein supplement needed in the elderly. The frequency of nutrition assessment in the elderly population is at least once a year; if there is a risk of malnutrition or nutrition problem, they need a reassessment three months later to ensure good nutrition condition. 

### 4.3. Methodological Considerations

There were several limitations to this study. Firstly, the number of available RCTs was insufficient, so that the statistical power was low because of the small study sample size. Secondly, studies were included from only three databases (PubMed, Embase, and Cochrane Library), which calls into question the generalization of the findings and the strength of the conclusions. Thirdly, the forms and dosage of protein supplements were not consistent. Finally, the total dietary protein intake was not considered, on the basis of which it can be assessed whether the quantity of total protein intake is sufficient or not. We suggest a further perspective study to explore the effect of the quantity of total protein intake, including dietary intake from food and supplements such as protein powder and protein drink, on sarcopenia.

## 5. Conclusions

Protein supplementation has no significant effect on improving skeletal muscle mass index, hand grip strength, 4 m gait speed, chair rise test, and SPPB. This may be caused by the lack of clinical standards on the form or dosage of protein supplements. Therefore, a large population cohort study needs to be established in the future for further discussion.

## Figures and Tables

**Figure 1 healthcare-09-00650-f001:**
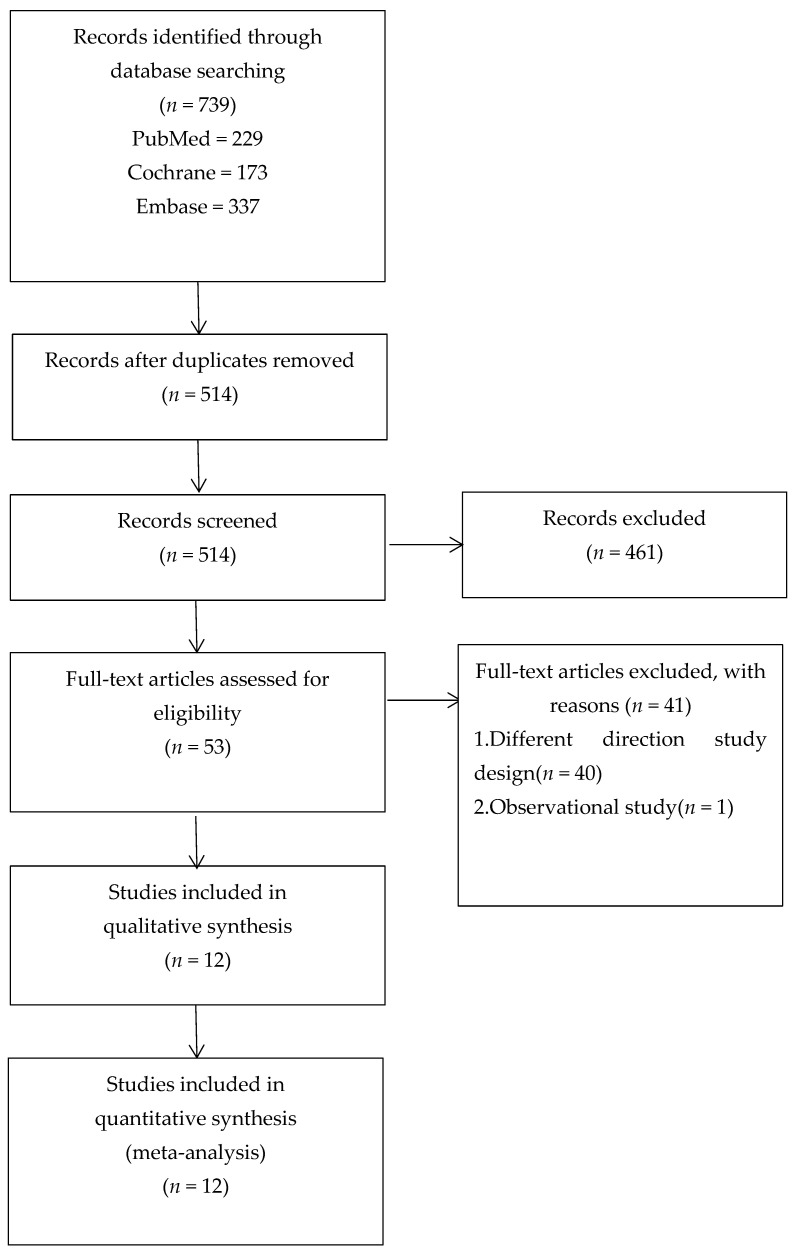
PRISMA flow diagram.

**Figure 2 healthcare-09-00650-f002:**
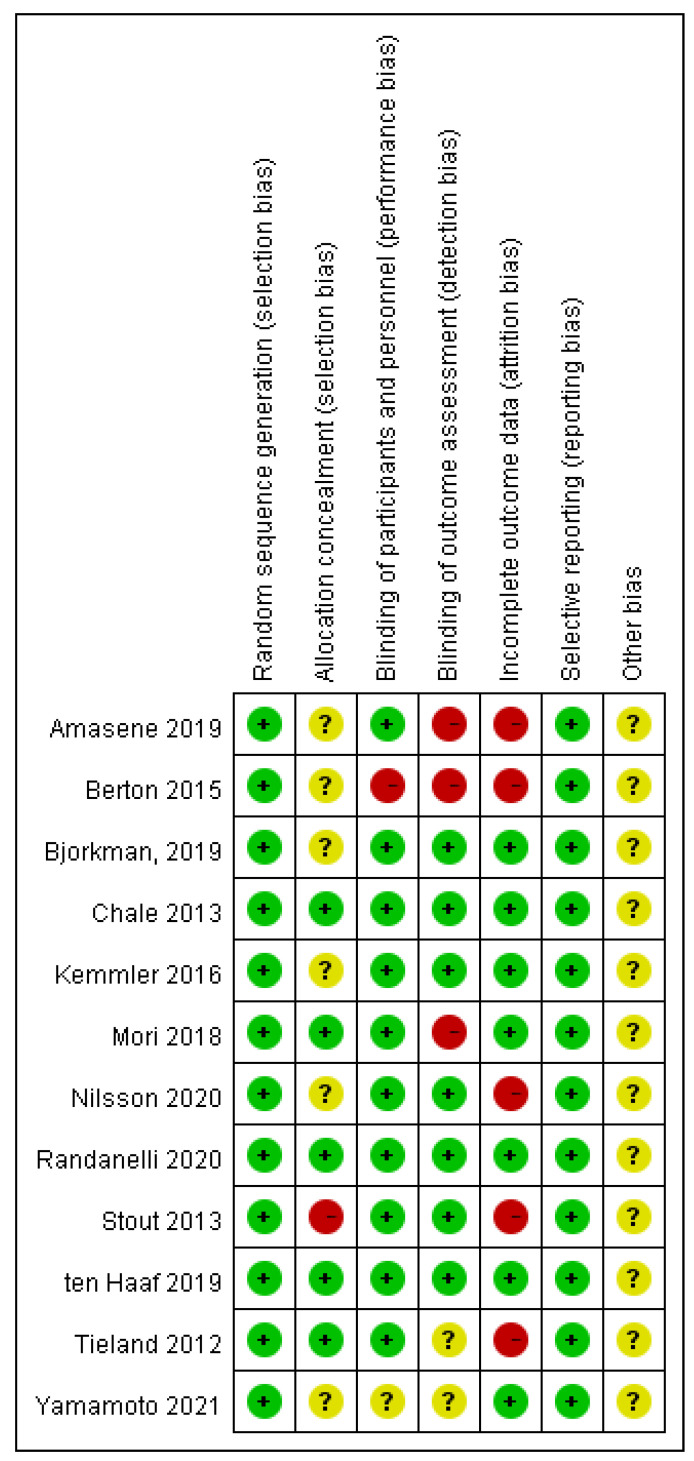
Risk-of-bias assessments for the selected studies.

**Figure 3 healthcare-09-00650-f003:**
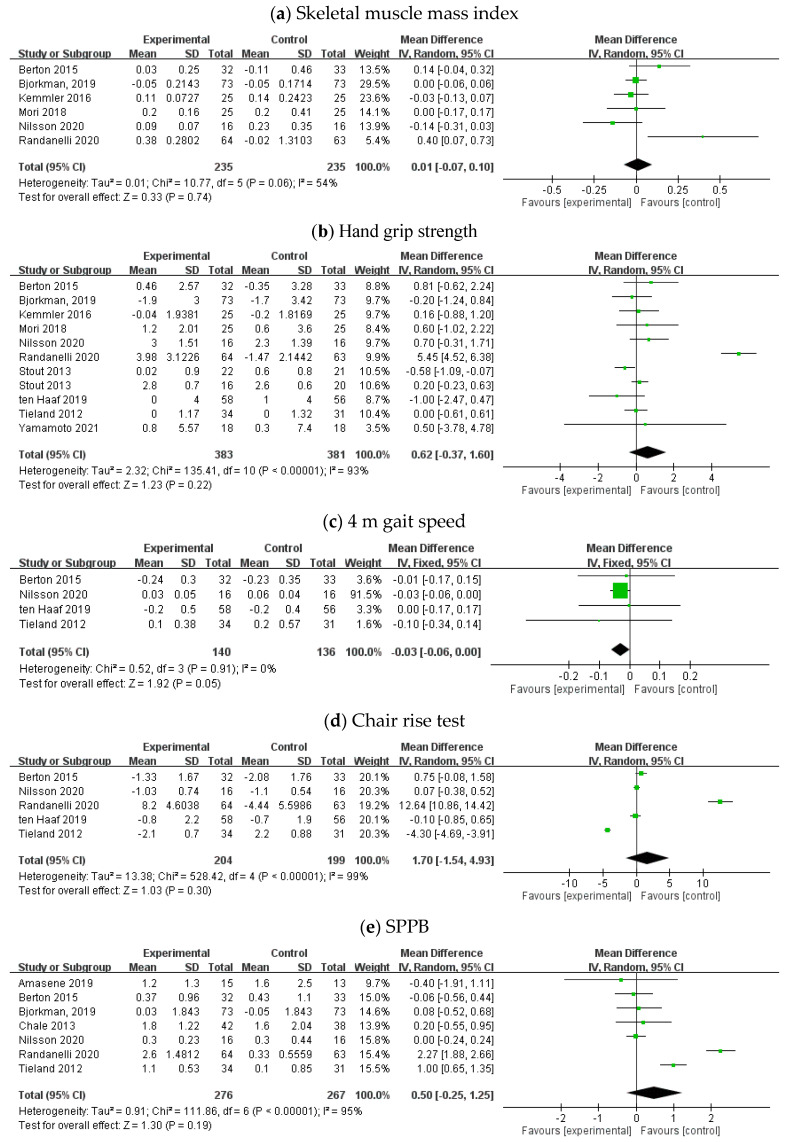
Forest plot of the effect of protein intake on (**a**) muscle mass: skeletal muscle mass index, (**b**) muscle strength: hand grip strength, physical performance: (**c**) 4 m gait speed, (**d**) chair rise test, and (**e**) short physical performance battery (SPPB) in randomized clinical trials.

**Figure 4 healthcare-09-00650-f004:**
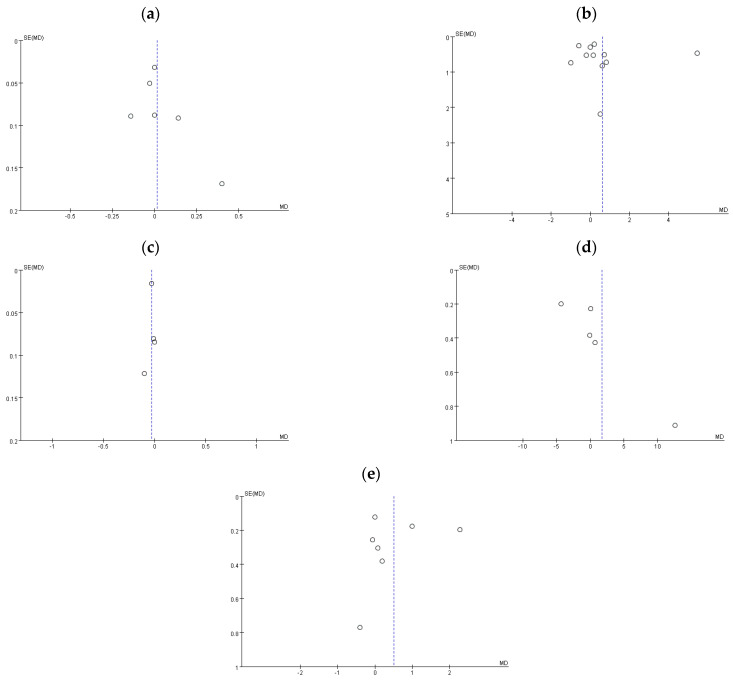
Funnel plot: (**a**) skeletal muscle mass index, (**b**) hand grip strength, (**c**) 4 m gait speed, (**d**) chair rise test, and (**e**) SPPB.

**Table 1 healthcare-09-00650-t001:** Search strategy in PubMed till 20 May 2021 (similar search run in Cochrane Library).

1. Elderly [MeSH Terms] OR [All Fields]
2. Elder [MeSH Terms] OR [All Fields]
3. Older [MeSH Terms] OR [All Fields]
4. Aged [MeSH Terms] OR [All Fields]
5. 1 OR 2 OR 3 OR 4
6. Sarcopenia [MeSH Terms] OR [All Fields]
7. 5 AND 6
8. Protein intake [MeSH Terms] OR [All Fields]
9. Leucine [MeSH Terms] OR [All Fields]
10. HMB [MeSH Terms] OR [All Fields]
11. β-hydroxy-β-methylbutyric acid [MeSH Terms] OR [All Fields]
12. β-hydroxy-β-methylbutyrate [MeSH Terms] OR [All Fields]
13. 8 OR 9 OR 10 OR 11 OR 12
14. Muscle mass [MeSH Terms] OR [All Fields]
15. Muscle strength [MeSH Terms] OR [All Fields]
16. Hand grip strength [MeSH Terms] OR [All Fields]
17. Gait speed [MeSH Terms] OR [All Fields]
18. SARC-F questionnaire [MeSH Terms] OR [All Fields]
19. Chair stand test [MeSH Terms] OR [All Fields]
20. Short physical performance battery [MeSH Terms] OR [All Fields]
21. Time up and go test [MeSH Terms] OR [All Fields]
22. 6 min walking test [MeSH Terms] OR [All Fields]
23. 14 OR 15 OR 16 OR 17 OR 18 OR 19 OR 20 OR 21 OR 22
24. Effect [MeSH Terms] OR [All Fields]
25. Effective [MeSH Terms] OR [All Fields]
26. Effectiveness [MeSH Terms] OR [All Fields]
27. Efficacy [MeSH Terms] OR [All Fields]
28. Improvement [MeSH Terms] OR [All Fields]
29. 24 OR 25 OR 26 OR 27 OR 28
30. 23 AND 29
31. 7 AND 13 AND 30

**Table 2 healthcare-09-00650-t002:** Characteristics of the randomized controlled trials included in this systematic review and meta-analysis.

Author,	Participants	Intervention	Comparison	Duration	Results
Year					
Tieland, 2012	65 frail elderly	15 g protein, bid	Placebo	24 wk	HG--, chair rise--, 4 m gait speed--, SPPB↑
Chale,	80 adult aged 70–85 years	40 g whey protein/day	Isocaloric control	6 mo	SPPB--
2013					
Stout, 2013	43 male or female aged ≥65 years	3 g HMB/day	Placebo	24 wk	HG--
	36 male or female aged ≥65 years	3 g HMB/day	Placebo	24 wk	HG↑
	with resistance exercise				
Berton, 2015	65 healthy women	1.5 g HMB/day	Control	8 wk	SMI--, HG--, chair stand--,
					4-m gait speed, SPPB↑
Kemmler, 2016	50 women aged ≥70 years with sarcopenic obesity	40 g protein/day	Control	6 mo	SMI--, HG--
Mori,	50 women aged 65–80 years	22.3 g protein/day	Control	24 wk	SMI--, HG--
2018					
Amasene, 2019	28 participants aged >70 years,	20 g whey protein/day	Placebo	12 wk	SPPB--
Bjorkman, 2019	146 older persons (>74 years of age) with sarcopenia	20 g protein, bid	Placebo	12 mo	SMI--, HG--, SPPB--
ten Haaf, 2019	114 adults aged 67–73 years	36.8 g milk protein	Placebo	12 wk	HG--, chair stand--, 4 m gait speed--,
Nilsson, 2020	32 male aged ≥65 years	24 g whey protein/day	Placebo	12 wk	SMI↑, HG↑, chair rise--, 4 m gait speed↑, SPPB--
Rondanelli,	127 adults aged ≥65 years with sarcopenia	20 g whey protein, b.i.d.	Placebo	8 wk	SMI--, HG↑, chair rise↑, SPPB↑
2020					
Yamamoto, 2021	36 patients aged 70–79 years with type 2 DM	3 g protein, b.i.d.	Control	48 wk	HG--

The sign “--” and “↑” mean “no significantly change” and “significantly increase”.

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
