# Peer review of "Sarcopenia among the Elderly Population: A Systematic Review and Meta-Analysis of Randomized Controlled Trials"

_healthcare, 2021, doi:10.3390/healthcare9060650_

Round 1

Reviewer 1 Report

This systematic review and meta-analysis was conducted to explore the effect of protein intake on prevention and improvement of sarcopenia. 

The work was conducted in a formally correct manner and the (few) studies on the subject were all taken into account. The discussion of the work can be improved. The authors should elaborate on the aspects of the excessive use of protein supplements, the possible damage to health and the ease with which the protein quota required for the prevention of sarcopenia is achieved through diet alone. It is also useful to look at the differences between plant and animal proteins and the risks associated with too much of the latter in the diet.

As the data are very clear, it would also be appropriate to change the title of the review, suggesting the futility of supplementation. 

Author Response

Reply to Reviewer 1

This systematic review and meta-analysis was conducted to explore the effect of protein intake on prevention and improvement of sarcopenia. 

The work was conducted in a formally correct manner and the (few) studies on the subject were all taken into account. The discussion of the work can be improved. The authors should elaborate on the aspects of the excessive use of protein supplements, the possible damage to health and the ease with which the protein quota required for the prevention of sarcopenia is achieved through diet alone. It is also useful to look at the differences between plant and animal proteins and the risks associated with too much of the latter in the diet.

As the data are very clear, it would also be appropriate to change the title of the review, suggesting the futility of supplementation. 

Ans. Thanks for the reviewer’s useful comments. We apologize for the inadequate descriptions. The descriptions have been corrected.

Reviewer 2 Report

Dear authors:

After review the article, some changes are proposed in order to consider.

King Regards

Round 2

Reviewer 2 Report

Dear Author

Accepted it

This manuscript is a resubmission of an earlier submission. The following is a list of the peer review reports and author responses from that submission.